# The Improved WNOFRFs Feature Extraction Method and Its Application to Quantitative Diagnosis for Cracked Rotor Systems

**DOI:** 10.3390/s22051936

**Published:** 2022-03-02

**Authors:** Haiying Liang, Chencheng Zhao, Yungao Chen, Yang Liu, Yulai Zhao

**Affiliations:** 1School of Mechanical Engineering & Automation, Northeastern University, Shenyang 110819, China; 2010092@stu.neu.edu.cn (H.L.); 2070279@stu.neu.edu.cn (C.Z.); 2070096@stu.neu.edu.cn (Y.C.); ylzhao@stumail.neu.edu.cn (Y.Z.); 2Key Laboratory of Vibration and Control of Aero-Propulsion System Ministry of Education, Northeastern University, Shenyang 110819, China

**Keywords:** cracked rotor, WNOFRFs, fault diagnosis, nonlinear systems

## Abstract

During its operation, a rotor system can be exposed to multiple faults, such as rub-impact, misalignment, cracks and unbalancing. When a crack fault occurs on the rotor shaft, the vibration response signals contain some nonlinear components that are considerably tougher to be extracted through some linear diagnosis methods. By combining the Nonlinear Output Frequency Response Functions weighted contribution rate (WNOFRFs) and Kullback–Leibler (KL) divergence, a novel fault diagnosis method of improved WNOFRFs is proposed. In this method, an index, improved optimal WNOFRFs (*IOW*), is defined to represent the nonlinearity of the faulty rotor system. This method has been tested through the finite element model of a cracked rotor system and then verified experimentally at the shaft crack detection test bench. The results from the simulation and experiment verified that the proposed method is applicable and effective for cracked rotor systems. The *IOW* indicator shows high sensitivity to crack faults and can comprehensively represent the nonlinear properties of the system. It can also quantitatively detect the crack fault, and the relationship between the values of *IOW* and the relative depth of the crack is approximately positively proportional. The proposed method can precisely and quantitatively diagnose crack faults in a rotor system.

## 1. Introduction

Rotor systems can be considered the most common and critical mechanical components in rotating machinery. They are widely used in various engineering fields, such as aerospace, transportation and electric power [1]. However, due to the more demanding requirements for the mechanical equipment, the structure of the rotor is more complicated, so it is prone to breakdowns in operation due to assembly errors or failure of its critical components [2,3]. Crack faults in rotor shafts are a common phenomenon that can lead to catastrophic failure and great economic loss if not identified in time [4,5]. Therefore, it is essential to detect and identify crack fault in rotor systems.

Vibration signals play an important role in reflecting the health status of the rotor system [6,7,8,9]. At present, many studies have focused on fault diagnosis methods for cracked rotor systems through vibration signals. R. Gradzki et al. [10] presented a rotor fault detection method through the auto-correlation and power spectral density functions of the rotor response signals and confirmed the sensitivity of this method, as well as its reliability. L. Xiang et al. [11] employed the orbit morphological characteristics to diagnose cracks in rotor systems. Its effectiveness at detecting crack information was verified through the dynamic model analysis and experimental study. Lu et al. [12] proposed a crack identification method for breathing crack identification of rotors based on the Kriging surrogate model by extracting the super-harmonic nonlinear characteristics. The results of the finite element method and experiments verify that this method is effective, accurate and robust for breathing crack identification in rotors. A. Hajnayeb et al. [13] created a feature space including the features of the largest Lyapunov exponent, approximate entropy and correlation dimension for fault diagnosis in rotors due to cracks. Saeed N. A. et al. [14] utilized the bifurcation diagram, Poincaré map, frequency spectrum and whirling orbit to investigate the Jeffcott rotor system. This strategy can not only inspect the existence of a crack but also predict the crack size based on the qualitative change of the system’s whirling motions when the crack size increases. Yang et al. [15] proposed the hybrid MVA method and used the cracked bench rotor to validate its effectiveness. These methods can identify the crack fault of the rotor system, but all of them cannot quantitatively describe the crack to detect its depth.

The Frequency Response Function (FRF) is the basic theory in the analysis of a linear system. The FRF represents the relationship of the input and output for a linear system in the frequency domain. The theory of the FRF has been widely used in engineering practice [16]. However, most engineering systems are complicated and nonlinear, so they cannot be simply described through the linear model. In the late 1950s, the Generalized Frequency Response Functions (GFRFs) were developed in the nonlinear case based on the theory of the Volterra series. The GFRFs were defined as the multidimensional Fourier transformations of the Volterra kernel functions. They are used to represent the nonlinear characteristics in the frequency domain [17]. However, the calculation of high-order GFRFs is quite tedious because it will involve a large amount of algebra or symbolic manipulations [18]. Therefore, the GFRFs actually do not have a broad range of applications in engineering practice.

To tackle the above problem, Lang put forward the Nonlinear Output Frequency Response Functions (NOFRFs) on the basis of the Volterra series [19]. NOFRFs are a significant extension of the FRF theory for linear systems to nonlinear situations. It reveals the rationale of nonlinear phenomena well and provides a certain theoretical basis for analyzing nonlinear systems in the frequency domain [20]. One of the most attractive points of the NOFRFs is its one-dimensional nature, which has many advantages, so it is extensively used in engineering practice to solve some difficult problems, such as in the field of structural health monitoring and fault diagnosis [21]. Peng et al. [22] used the NOFRFs to detect cracks in beams, as well as the nonlinear component’s position of periodic structures [23]. Based on the Nonlinear Auto Regressive with eXegenous Input (NARX) model and the NOFRFs, Bayma et al. [24] designed a structural damage detection method for detecting cracks in beam structures. By introducing the evidence theory to the NOFRFs, Cao et al. [25] put forward a fault diagnosis method for complicated nonlinear systems by combining the NOFRFs and evidence theory. This approach was employed to diagnose the fault of the transmission system of a numerical control machine tool. Zhao et al. [26] put forward a method for detecting early damage to nonlinear systems by combining the NOFRFs with the convolutional neural network and long short-term memory network (CNN-LSTM) model, which was verified through a cantilever beam with a breathing crack.

Some studies are devoted to NOFRF-based indexes to enhance the effect of the NOFRFs for defect detection and fault diagnosis. Peng et al. [27] put forward an index, *Fe*(*n*), toward the Nonlinear Auto-Regressive Moving Average with eXegenous Inputs (NARMAX) model in damage detection. The experiments of aluminum plates with holes and cracks indicated that the index *Fe*(*n*) of the inspected structure is considerably different from the damage-free structure. Huang [28] put forward the index *Ne*, which integrates all orders of the NOFRFs by introducing information entropy. The experiments of the plate specimens were conducted to reveal the higher sensitivity of the index *Ne*. Mao et al. [29] introduced Kullback–Leibler (KL) divergence to the NOFRFs to propose the index NOFRFs-KL (*NKL*) for fatigue damage identification. The effectiveness of *NKL* was verified by an experiment on a train wheelset. For improving the sensitivity of the *Fe* and *Ne* indexes, Liu et al. [30,31] put forward the method of the Nonlinear Output Frequency Response Functions weighted contribution rate (WNOFRFs). The index of the second-order optimal weighted contribution rate *Rm* was defined for feature extraction of the rub-impact of the rotor system. Another index, *K*, which is based on Rm, was also defined. Its superiority was verified through the mathematical model and experiments on a cracked rotor system. Li et al. [32] proposed the index *J* by introducing the Clenshaw–Curtis numerical integration method and the minimum mutual entropy principle to the WNOFRFs for diagnosing the rotor rub-impact.

Further studies on the feature extraction method of the WNOFRFs and the associated index K indicated that this method can improve the performance of crack detection. However, in this method, only the second-order WNOFRFs are focused on, rather than comprehensively considering the other orders of the WNOFRFs. The increments of the index *K* are a bit small when the degree of the crack fault increases, so it is not sensitive enough for crack fault detection. The values of *K* and the degree of the crack are in a linear relationship to some extent rather than in a positive proportion, as might be expected. In the present study, a fault diagnosis method known as the improved WNOFRFs is proposed, which is employed to diagnose rotor cracks. An associated index denoted as *IOW* is proposed to represent the nonlinear characteristics of the system. The effectiveness of the method was verified through a mathematical model of the cracked Jeffcott rotor system and an experimental rotor system with crack faults.

## 2. The Principle of Improved WNOFRFs

### 2.1. NOFRFs Theory

For a stable linear system, its output frequency response can be expressed as
(1)Y(jω)=H(jω)U(jω),
where U(jω) and Y(jω) are the input and output spectra, respectively, which are the Fourier transforms of the input function u(t) and the output function y(t) of the system. For linear systems, this expression shows that the possible output frequencies are the same as the frequencies in the input.

Considering the relationship in the time domain of the input and the output of a nonlinear system which are stable at zero equilibrium, this can be represented by the Volterra series:(2)$$y(t)=\sum_{n=1}^{+\infty }y_{n}(t)=\sum_{n=1}^{N}\int_{-\infty }^{\infty }\cdots \int_{-\infty }^{\infty }h_{n}\left(\tau_{1},\ldots ,\tau_{{}_{n}}\right)\prod_{i=1}^{n}u\left(t-\tau_{i}\right)d\tau_{i},$$
where u(t) denotes the input function of the nonlinear system at discrete time *t*, y(t) is the corresponding output function, $h_{n}(\tau_{1},\cdots ,\tau_{n})$ is the *n*th discrete time Volterra kernel, *N* denotes the maximum order of the system nonlinearity and τ is the time delay.

The output frequency response of the nonlinear system under general excitation can be represented by
(3)$$Y(j\omega )=\sum_{n=1}^{N}Y_{n}(j\omega ),$$
where Y(jω) represents the out spectrum of the system and $Y_{n}(j\omega )$ represents the *n*th order output frequency response of the nonlinear system, which is defined as
(4)$$Y_{n}(j\omega )=\frac{1/\sqrt{n}}{{\left(2\pi \right)}^{n-1}}\times \int_{\omega_{1}+\cdot \cdot \cdot +\omega_{n}=\omega }H_{n}(j\omega_{1},\cdot \cdot \cdot ,j\omega_{n})\prod_{i=1}^{n}U(j\omega_{i})d\sigma_{n\omega },$$

Equation (4) is a natural extension of the well-known linear relationship (expressed in Equation (1)) to non-linear cases, which is much more complicated than in linear system cases. It reveals how the nonlinear mechanisms operate on the input spectra to produce the system output frequency response. $\int_{\omega_{1}+\cdot \cdot \cdot +\omega_{n}=\omega }H_{n}(j\omega_{1},\cdot \cdot \cdot ,j\omega_{n})\prod_{i=1}^{n}U(j\omega_{i})d\sigma_{n\omega }$ is the integral of $H_{n}(j\omega_{1},\cdots ,j\omega_{n})\prod_{i=1}^{n}U(j\omega_{i})$ under the condition of $\omega_{1}+\cdots +\omega_{n}=\omega$ in the n-dimensional hyperplane. $H_{n}(j\omega_{1},\cdots ,j\omega_{n})$ is the Fourier transform of $h_{n}(j\omega_{1},\cdots ,j\omega_{n})$ and denotes the *n*th GFRFs of the nonlinear systems [33]. Its expression is
(5)$$H_{n}\left(j\omega_{1},\ldots ,j\omega_{n}\right)=\int_{-\infty }^{\infty }\ldots \int_{-\infty }^{\infty }h_{n}\left(\tau_{1},\ldots ,\tau_{n}\right)\times e^{-\left(\omega_{1}\tau_{1}+\cdots +\omega_{n}\tau_{n}\right)j}d\tau_{1}\ldots d\tau_{n},$$

Under the condition of $U_{n}(j\omega )=\int_{\omega_{1}+\cdots +\omega_{n}=\omega }\prod_{i=1}^{n}U(j\omega_{i})d\sigma_{n\omega }\neq 0$, the *n*th NOFRFs is defined as
(6)$$G_{n}(j\omega )=\frac{\int_{\omega_{1}+\cdots +\omega_{n}=\omega }H_{n}(j\omega_{1},\cdots ,j\omega_{n})\prod_{i=1}^{n}U(j\omega_{i})d\sigma_{n\omega }}{\int_{\omega_{1}+\cdots +\omega_{n}=\omega }\prod_{i=1}^{n}U(j\omega_{i})d\sigma_{n\omega }},$$

This is substituted into Equation (4), and the *n*th output frequency response of the nonlinear system can be rewritten as
(7)$$\begin{array}{l} Y_{n}(j\omega )=\frac{\int_{\omega_{1}+\cdots +\omega_{n}=\omega }H_{n}(j\omega_{1},\cdots ,j\omega_{n})\prod_{i=1}^{n}U(j\omega_{i})d\sigma_{n\omega }}{\int_{\omega_{1}+\cdots +\omega_{n}=\omega }\prod_{i=1}^{n}U(j\omega_{i})d\sigma_{n\omega }}\times \frac{1/\sqrt{n}}{{\left(2\pi \right)}^{n-1}}\prod_{i=1}^{n}U(j\omega_{i})d\sigma_{n\omega }\\ \begin{array}{cc} & = \end{array}G_{n}(j\omega )U_{n}(j\omega ) \end{array}$$

Therefore, the relationship between the output spectrum and the input spectrum can be rewritten as [34]
(8)$$Y(j\omega )=\sum_{n=1}^{N}Y_{n}(j\omega )=\sum_{n=1}^{N}G_{n}(j\omega )U_{n}(j\omega ),$$

The first four-order NOFRFs are generally sufficient to reflect the nonlinear characteristics of a nonlinear system, so only the first four-order NOFRFs are considered.

Therefore, Equation (8) can be further expressed as [30]
(9)$$Y(j\omega )=\sum_{n=1}^{N}Y_{n}(j\omega )=\sum_{n=1}^{N}G_{n}(j\omega )U_{n}(j\omega ),$$
(10)$$Y(j2\omega )=G_{2}(j2\omega )U_{2}(j2\omega )+G_{4}(j2\omega )U_{4}(j2\omega ),$$
(11)$$Y(j3\omega )=G_{3}(j3\omega )U_{3}(j3\omega ),$$
(12)$$Y(j4\omega )=G_{4}(j4\omega )U_{4}(j4\omega ),$$

As shown in Equations (9)–(12), if a nonlinear system was excited twice by two harmonic input signals that had different amplitudes, the first four-order NOFRFs could be calculated through the least square algorithm.

The Fourier transforms of these two harmonic inputs are $A_{i}^{\left(1\right)}(j\omega_{F})$ and $A_{i}^{\left(2\right)}(j\omega_{F})$, i=1,2,3,4. The output spectra are expressed as
(13)$$\left(\begin{array}{c} Y^{\left(1\right)}(j\omega_{F})\\ Y^{\left(2\right)}(j\omega_{F}) \end{array}\right)=\left(\begin{array}{cc} A_{1}^{\left(1\right)}(j\omega_{F}) & A_{3}^{\left(1\right)}(j\omega_{F})\\ A_{1}^{\left(2\right)}(j\omega_{F}) & A_{3}^{\left(2\right)}(j\omega_{F}) \end{array}\right)\left(\begin{array}{c} G_{{}^{1}}(j\omega_{F})\\ G_{{}^{3}}(j\omega_{F}) \end{array}\right),$$

$G_{1}(j\omega_{F})$ and $G_{3}(j\omega_{F})$ can be calculated as follows:(14)$$\left(\begin{array}{c} G_{1}(j\omega_{F})\\ G_{3}(j\omega_{F}) \end{array}\right)={\left(\begin{array}{cc} A_{1}^{\left(1\right)}(j\omega_{F}) & A_{3}^{\left(1\right)}(j\omega_{F})\\ A_{1}^{\left(2\right)}(j\omega_{F}) & A_{3}^{\left(2\right)}(j\omega_{F}) \end{array}\right)}^{-1}\left(\begin{array}{c} Y^{\left(1\right)}(j\omega_{F})\\ Y^{\left(2\right)}(j\omega_{F}) \end{array}\right),$$

The remaining values of the NOFRFs can be calculated in a similar way.

### 2.2. The Principle of WNOFRFs

On the basis of the associated index Fe proposed by Peng [27], Liu [30,31] introduced the weighting coefficient $n^{\rho }$ and proposed the feature extraction method known as the NOFRFs weighted contribution rate (WNOFRFs). Its expression is
(15)$$Rn(n)(\rho )=\frac{\int_{-\infty }^{+\infty }\;\left|\frac{G_{n}(j\omega )}{n^{\rho }}\right|d\omega }{\sum_{i=1}^{N}\int_{-\infty }^{+\infty }\left|\frac{G_{i}(j\omega )}{i^{\rho }}\right|d\omega },\;1\leq n\leq N,\;\rho \in (-\infty ,0),$$
where the weighting coefficient $n^{\rho }$ amplifies the high-order $G_{n}(j\omega )$ of the nonlinear system.

Further studies found that the second-order weighted contribution rate Rn2(ρ) is a good index for indicating the degree of the fault, and Rn2(ρ) is defined as
(16)$$Rn2(\rho )=\frac{\int_{-\infty }^{+\infty }\left|\frac{G_{2}(j\omega )}{2^{\rho }}\right|d\omega }{\sum_{i=1}^{N}\int_{-\infty }^{+\infty }\left|\frac{G_{i}(j\omega )}{i^{\rho }}\right|d\omega }\;\;\;\;\rho \in (-\infty ,0),$$

In Equation (16), the maximum value of the curve Rn2(ρ) was denoted as the second-order optimal WNOFRFs *Rm*, and the Newton-Raphson iterative method was applied to solve the *x*-coordinate corresponding to *Rm*, which is named as the optimal fitness factor $\rho_{\max}$. Its expression is as follows:(17)$$Rm=\frac{\int_{-\infty }^{+\infty }\left|\frac{G_{2}(j\omega )}{2^{\rho_{\max}}}\right|d\omega }{\sum_{i=1}^{N}\int_{-\infty }^{+\infty }\left|\frac{G_{i}(j\omega )}{i^{\rho_{\max}}}\right|d\omega }\;,$$

On the basis of *Rm*, an associated index *K* was defined for solving the problem that the sensitivity of *Rm* changes significantly at different stages of crack growth [31]. The index *K* can be calculated through
(18)$$K=Rm^{1/e},$$

This index is used to represent the system’s nonlinearity, and the severity of the faults can be distinguished by *Rm* to some extent. However, for the index *K*, only the second-order WNOFRFs are focused on, rather than comprehensively considering the other orders of the WNOFRFs. The increments of *K* are a bit small when the degree of the crack fault increases, so it is not sensitive enough to crack faults, and the values of *K* and the degree of the crack are in a linear relationship to some extent rather than in a positive proportion, as might be expected.

### 2.3. Improved WNOFRFs Fault Diagnosis Method

To explore an index that has a high enough sensitivity and its values having a certain relationship with the degree of faults to precisely and quantitatively diagnose faults, KL divergence is combined with WNOFRFs.

The matching degree between the referenced probability density distribution $f(x_{i}),i=1,\cdots ,N$ and the target probability density distribution $g(x_{i}),i=1,\cdots ,N$ can be measured by calculating their KL divergence. If the difference between $f(x_{i})$ and $g(x_{i})$ is larger, their KL divergence $D_{KL}(f∥g)$ will be larger [29]. The expression of the KL divergence is given by
(19)$$D_{KL}(f∥g)=\sum_{i=1}^{N}f(x_{i})\log\frac{f(x_{i})}{g(x_{i})},$$

According to Equation (19), the discrepancy between two WNOFRFs, such as the WNOFRF of the healthy rotor system and the WNOFRF of the cracked rotor system, can be represented through calculating the KL divergence. Therefore, to explore a superior fault diagnosis method, a fault diagnosis method of the improved WNOFRFs is proposed based on the WNOFRFs and the KL divergence. Its expression is as follows:(20)$$IW(\rho )=IW1(\rho )+IW2(\rho )+IW3(\rho )+IW4(\rho )=\sum_{n=1}^{4}\left|Rn_{h}(n)(\rho )\log\frac{Rn_{h}(n)(\rho )}{Rn_{t}(n)(\rho )}\right|,n=1,2,3,4,$$

In Equation (20), $Rn_{h}(n)(\rho )$ represents the *n*th-order WNOFRFs of the healthy rotor system and $Rn_{t}(n)(\rho )$ represents the *n*th-order WNOFRFs of the target rotor system.

If the target rotor system does not have any crack faults, the curve $Rn_{t}(n)(\rho )$ approximately coincides with $Rn_{h}(n)(\rho )$, so for a specific $\rho =\rho_{0}$, $\log\frac{Rn_{h}(n)(\rho_{0})}{Rn_{t}(n)(\rho_{0})}$ is near zero. At the same time, the value of $IW(\rho_{0})$ is close to zero. If the target rotor system contains crack faults, $Rn_{t}(n)(\rho_{0})$ is bigger than $Rn_{h}(n)(\rho )$ for a specific $\rho =\rho_{0}$, so $\log\frac{Rn_{h}(n)(\rho_{0})}{Rn_{t}(n)(\rho_{0})}<0$. When the crack fault gets worse, the value of $Rn_{t}(n)(\rho_{0})$ increases, so for a specific $\rho =\rho_{0}$, the value of $\log\frac{Rn_{h}(n)(\rho_{0})}{Rn_{t}(n)(\rho_{0})}$ decreases while the value of $\left|Rn_{h}(n)(\rho_{0})\log\frac{Rn_{h}(n)(\rho_{0})}{Rn_{t}(n)(\rho_{0})}\right|$ increases. Therefore, the value of $IW(\rho_{0})$ gradually increases as the crack fault gets worse.

## 3. Mathematical Model of the Cracked Rotor System

The mathematical model in this study was considered to employ the Jeffcott rotor system [35] with a crack fault, which is shown in Figure 1.

The rotor shaft contains 11 shaft units, each of which has 2 nodes. The specific parameters of the units are listed in Table 1.

The material of the rotor was 45#steel, the properties of which are as follows. The Young’s modulus is 210 GPa, the density is 7850 kg/m^3^, Poisson’s ratio is 0.3, the bearing stiffness is 2 × 10^6^ and the unbalancing is 167.4 × 10^−6^ kg·m and 236.7 × 10^−6^ kg·m.

Each shaft segment of the rotor system is modeled by the Timoshenko beam. Each shaft segment contains two nodes. Four freedoms of each node are taken into account, and the displacement vector of the *i*th node is defined as
(21)$$q_{i}={\left[\begin{array}{cccc} x_{i} & y_{i} & \theta_{xi} & \theta_{yi} \end{array}\right]}^{T},$$

The dynamic equation of the Jeffcott rotor system can be expressed as
(22)$$M\ddot{q}+(C+G)\dot{q}+K^{e}q=F,$$
where ***M*** is the mass matrix, ***C*** is the damping matrix, ***G*** is the gyro matrix, ***K*** is the stiffness matrix and ***F*** is the excitation vector.

There is a crack in a certain shaft segment, the radius of which is *R* and the length of which is *L*. The cracked shaft element is illustrated in Figure 2. Figure 2a is a diagram of the cracked shaft element. Figure 2b is the cross-sectional view of the crack. In Figure 2a, the element is loaded with axial forces *P*_3_ and *P*_9_, shear forces *P*_1_, *P*_2_, *P*_7_ and *P*_8_, bending moments *P*_4_, *P*_5_, *P*_10_ and *P*_11_ and torques *P*_6_ and *P*_12_. In Figure 2b, *a* represents the depth of the crack, and *b* is its width. The relative depth of the crack is represented as *a*/*R*.

The stiffness matrix of the crack-free shaft element is
(23)$$K_{u}^{e}=T\left[\begin{array}{cccc} L^{3}/3\mathrm{EI} & 0 & 0 & L^{2}/2\mathrm{EI}\\ 0 & L^{3}/3\mathrm{EI} & -L^{2}/2\mathrm{EI} & 0\\ 0 & L^{2}/2\mathrm{EI} & L/\mathrm{EI} & 0\\ L^{2}/2\mathrm{EI} & 0 & 0 & L/\mathrm{EI} \end{array}\right]T^{T},$$
where the transition matrix $T=\left[\begin{array}{cccccccc} -1 & 0 & 0 & -L & 1 & 0 & 0 & 0\\ 0 & -1 & L & 0 & 0 & 1 & 0 & 0\\ 0 & 0 & -1 & 0 & 0 & 0 & 1 & 0\\ 0 & 0 & 0 & -1 & 0 & 0 & 0 & 1 \end{array}\right]$.

According to the Castigliano theorem, the general strain energy of the cracked shaft element is expressed as
(24)$$U=U^{0}+U^{c},$$
where *U*^0^ is the strain energy of the crack-free shaft element and *U^c^* is the strain energy of the cracked shaft element.

Based on the fracture mechanics, *U^c^* can be expressed as
(25)$$U^{c}=\int_{A}J(A)dA,$$
where *J*(*A*) is the strain energy function.

According to the Paris theory, the flexibility of the cracked shaft element caused by *P_i_* can be expressed as
(26)$$u_{i}=\frac{\partial U}{\partial P_{i}}=\frac{\partial U^{0}}{\partial P_{i}}+\frac{\partial U^{c}}{\partial P_{i}}=u_{i}^{0}+u_{i}^{c},$$
where *u*^0^ represents the flexibility of the crack-free shaft element and *u^c^* represents the additional flexibility due to cracking [36].

The flexibility coefficients of the cracked shaft element can be expressed as
(27)$$c_{ij}=\frac{\partial u_{i}}{\partial P_{j}}=\frac{\partial u_{i}^{0}}{\partial P_{j}}+\frac{\partial u_{i}^{c}}{\partial P_{j}}=\frac{\partial^{2}}{\partial P_{i}P_{j}}\int_{A}U^{0}dA+\frac{\partial^{2}}{\partial P_{i}P_{j}}\int_{A}U^{c}dA,$$

Under the assumption that the direction of the crack is on the same side as the mass eccentricity, only the mixed situation of a Mode Ι crack caused by the bending moment *P*_4_ and a Mode ΙΙ crack caused by the shear force *P*_2_ is considered, as the direction of *P*_2_ and *P*_4_ is the same as the direction of the crack. Based on this, the strain energy of the crack-free shaft element is expressed as
(28)$$U^{0}=\int_{0}^{L}\frac{(P_{4}+P_{2})x}{2\mathrm{EI}}d\alpha =\frac{1}{2\mathrm{EI}}\left(P_{4}^{2}L+P_{4}P_{2}L^{2}+\frac{P_{2}^{2}L^{3}}{3}\right),$$

Thus, the flexibility matrix of the crack-free shaft element is given as
(29)$$c^{0}=\frac{1}{\mathrm{EI}}\left[\begin{array}{cc} L^{3}/3 & L^{2}/2\\ L^{2}/2 & L \end{array}\right],$$

The stress intensity factors $K_{Ι4}$ and $K_{ΙΙ4}$, corresponding to the bending moment *P*_4_ and the shear force *P*_2_, are defined as
(30)$$K_{I4}=\sigma_{4I}\sqrt{\pi z}F_{I}(\frac{z}{h}),$$
(31)$$K_{\mathrm{II}2}=\sigma_{2\mathrm{II}}\sqrt{\pi z}F_{\mathrm{II}}(\frac{z}{h}),$$
where $\sigma_{4I}=\frac{2(P_{2}\alpha -P_{4})h}{\pi R^{4}}$, $\sigma_{2\mathrm{II}}=\frac{P_{2}}{\pi R^{2}}$, $F_{I}(\frac{z}{h})=\sqrt{\frac{2h}{\pi z}\tan\frac{\pi z}{2h}}\frac{0.923+0.199{\left[1-\sin(\pi z/2h)\right]}^{4}}{\cos(\pi z/2h)}$, $F_{\mathrm{II}}(\frac{z}{h})=\frac{1.122-0.561(z/h)+0.085{\left(z/h\right)}^{2}+0.18{\left(z/h\right)}^{3}}{(1-z/h{)}^{1/2}}$ and $h=2\sqrt{R^{2}-w^{2}}$.

Therefore, the additional strain energy caused by the crack with depth *a* and width *b* in Figure 2b is
(32)$$U^{c}=\frac{1-v^{2}}{E}\int_{-b}^{b}dw\int_{0}^{\sqrt{R^{2}-w^{2}}-(R-a)}\left[\frac{4(P_{2}^{2}\alpha^{2}-2P_{2}P_{4}\alpha +P_{4}^{2})h^{2}}{\pi R^{8}}zF_{I}^{2}\left(\frac{z}{h}\right)+\frac{P_{2}^{2}}{\pi^{2}R^{4}}\pi zF_{\mathrm{II}}^{2}\left(\frac{z}{h}\right)\right]dz,$$

Thus, the local additional flexibility coefficients *c*_44_ and *c*_22_ caused by the bending moment *P*_4_ and the shear force *P*_2_ are expressed as
(33)$$c_{44}=\frac{\partial^{2}U^{c}}{\partial P_{4}^{2}}=\frac{1-v^{2}}{E}\int_{-b}^{b}\frac{32{\left(R^{2}-w^{2}\right)}^{2}}{\pi R^{8}}dw\int_{0}^{\left[\sqrt{R^{2}-w^{2}}-(R-a)\right]/h}\frac{z}{h}F_{I}^{2}\left(\frac{z}{h}\right)d\left(\frac{z}{h}\right),$$
(34)$$\begin{array}{c} c_{22}=\frac{\partial^{2}U^{c}}{\partial P_{2}^{2}}=\frac{1-v^{2}}{E}\left[\int_{-b}^{b}\frac{128\alpha^{2}{\left(R^{2}-w^{2}\right)}^{2}}{\pi R^{8}}dw\int_{0}^{\left[\sqrt{R^{2}-w^{2}}-(R-a)\right]/h}\frac{z}{h}F_{I}^{2}\left(\frac{z}{h}\right)d\left(\frac{z}{h}\right)\right.\\ +\left.\int_{-b}^{b}\frac{8{\left(R^{2}-w^{2}\right)}^{2}}{\pi R^{4}}dw\int_{0}^{\left[\sqrt{R^{2}-w^{2}}-(R-a)\right]/h}\frac{z}{h}F_{\mathrm{II}}^{2}\left(\frac{z}{h}\right)d\left(\frac{z}{h}\right)\right] \end{array},$$

The additional flexibility matrix caused by the crack is given by
(35)$$c^{c}=\left[\begin{array}{cc} c_{22} & 0\\ 0 & c_{44} \end{array}\right],$$

Thus, the flexibility matrix of the full-open crack is expressed as
(36)$$c=c^{0}+c^{c}=\frac{1}{\mathrm{EI}}\left[\begin{array}{cc} L^{3}/3 & L^{2}/2\\ L^{2}/2 & L \end{array}\right]+\left[\begin{array}{cc} c_{22} & 0\\ 0 & c_{44} \end{array}\right],$$

The stiffness matrix of the full-open crack is
(37)$$K_{c}^{e}=T_{1}^{T}c^{-1}T_{1},$$
where the transition matrix $T_{1}=\left[\begin{array}{cccc} -1 & -L & 1 & 0\\ 0 & -1 & 0 & 1 \end{array}\right]$.

The flexibility matrix of the semi-open crack is expressed as
(38)$$c_{h}=c^{0}+\frac{1}{2}c^{c},$$

The stiffness matrix of the semi-open crack is
(39)$$K_{h}^{e}=T_{1}^{T}c_{h}^{-1}T_{1},$$

The stiffness matrix of the cracked shaft element varies with the time rotation angles due to the opening and closing characteristics of the crack. This variation [37] is expressed as
(40)$$K=K_{0}+K_{1}\cos(wt)+K_{2}\cos(2wt)+K_{3}\cos(3wt)+K_{4}\cos(4wt),$$
where ***K***_4_
$\left\{\begin{array}{l} \begin{array}{l} K_{0}=(5K_{u}^{e}+5K_{c}^{e}+6K_{h}^{e})/16\\ K_{1}=(9K_{u}^{e}-9K_{c}^{e})/16 \end{array}\\ K_{2}=(K_{u}^{e}+K_{c}^{e}-2K_{h}^{e})/4\\ K_{3}=(K_{c}^{e}-K_{u}^{e})/16\\ K_{0}=(-K_{u}^{e}-K_{c}^{e}+2K_{h}^{e})/16 \end{array}\right.$.

The Newmark-β algorithm is employed to solve the dynamic equation of the Jeffcott rotor system at different crack faults. The rotor system is excited twice using different unbalanced forces. The depth of the cracks is set to 1 mm, 2 mm, 3 mm or 4 mm, so the relative depth (representing the ratio between the crack depth and the radius of the shaft) is 0, 0.2, 0.4, 0.6 or 0.8, and the dynamic responses at node 8 of the crack-free rotor system and cracked rotor systems with different degrees of crack faults at the rotation speed of 1600 rpm are shown in Figure 3. The time domain responses are shown in Figure 3a, and the frequency domain responses of the mathematical model are shown in Figure 3b. It can be observed from Figure 3a that the domain response of the crack-free rotor system was a standard sine signal. With the degree of the crack fault getting worse (the relative depth of the crack ranges from 0 to 0.8), the amplitude of the time domain responses gradually increased both at the peak and at the trough. This change was not distinct at the peak but visible at the trough. As crack faults became gradually more serious, the peak values of the 1× component in the frequency domain response increased from 0.0257 mm to 0.02721 mm (0.0257 mm, 0.02579 mm, 0.02596 mm, 0.02596 mm and 0.02721 mm). The change was not visible, and the peak values of the 2× component in the frequency domain response increased from 6.325 × 10^−6^ mm to 1.196 × 10^−3^ mm (6.325 × 10^−6^ mm, 3.54 × 10^−5^ mm, 1.226 × 10^−4^ mm, 3.672 × 10^−4^ mm and 1.196 × 10^−3^ mm). The relative change was distinct. In Figure 3b, it is found that the 3×, 4× and 5× components also appeared in the condition of the most serious crack fault, which was due to changes in the local flexural stiffness of the cracked rotor [38]. However, these changes were insufficient to recognize the depth of the cracks through simple spectral analysis and some linear methods. Therefore, a novel nonlinear method needs to be explored to evaluate the cracked rotor system.

## 4. Fault Diagnosis of the Cracked Rotor System

### 4.1. Improved WNOFRFs for the Cracked Rotor System

Dynamic responses from the mathematical model of the cracked Jeffcott rotor system under different severities of the crack faults and corresponding excitations were used for the fault diagnosis through the method of improved WNOFRFs. First, the WNOFRFs were calculated according to Equation (15). The curves of the WNOFRFs (*Rn*) under different crack faults are shown in Figure 4.

It can be found from Figure 4 that the changing trends of the first four orders of WNOFRFs were relatively distinct. *Rn*1 monotonically increased from 0 to 1 with the increase in *ρ*. By contrast, *Rn*4 monotonically decreased from 1 to 0. When *ρ* was constant, the values of *Rn*1 only slightly decreased as the crack depth became larger, while *Rn*4 only had very small changes. Figure 4b indicates that *Rn*2 first rose from 0 to the maximum, which corresponds with the optimal fitness factor. Then, it declined to zero. The maximum values of *Rn*2 under different crack fault severities increased from zero, with the crack fault getting worse. As shown in Figure 4c, *Rn*3 followed the same trend as *Rn*2. However, its maximum values manifested a reverse trend as the relative depths of the crack faults deepened.

Next, *IW*1–*IW*4 were further obtained according to Equation (20). The curves of *IW*1–*IW*4 under different crack depths are presented in Figure 5.

It can be seen from Figure 5 that the changing trend of *IW*1, *IW*2, *IW*3 and *IW*4 was similar to that of *Rn*2. The maximum values of each order of improved WNOFRFs under different conditions were also similar to that of *Rn*2.

To integrate all orders of WNOFRFs, *IW*1, *IW*2, *IW*3 and *IW*4 were summed to yield the *IW*, the curves of which are shown in Figure 6. It can be obviously observed from Figure 6 that the changing trend of *IW* was first from 0 to the maximum. Then, it declined to zero. The maximum values of *IW* under different relative depths of crack faults increased from zero, with the fault getting worse.

### 4.2. The Index of IOW for the Cracked Rotor System

In this subsection, a novel index representing the nonlinear characteristics of the system, the improved optimal WNOFRFs (*IOW*), is proposed. Its expression is
(41)$$IOW=\max KLRn(\rho )=KLRn(\rho_{\mathrm{opt}}),$$
where $\rho_{\mathrm{opt}}$ represents the optimal fitness factor. In Figure 6, when $\rho =\rho_{\mathrm{opt}}$, the corresponding maximum value of every curve is denoted as *IOW*.

In this paper, the index *IOW* was used to detect the cracked rotor system. The *IOW* and *K* of the finite element model under different crack conditions were calculated according to Equations (41) and (18). The values of *IOW* and *K* are shown in Figure 7.

From Figure 7, a conspicuous increasing trend could be found for the values of *IOW* and *K* (*IOW* was from 0 to 3.410, and *K* was from 0.160 to 0.604), with the relative depth of the cracks increasing. These two indexes both can recognize crack faults in the rotor system. When comparing these two indexes, the values of *IOW* were larger than *K* under the same degree of cracking, so the index *IOW* was more sensitive.

Then, we explored the relationship between the values of the indexes and cracks. The relative depths were set from 0 to 0.8 in increments of 0.01. The values of the indexes were calculated, and then linear fitting was performed on the values. The fitting results of indexes *IOW* and *K* are illustrated in Figure 8 and Figure 9. Figure 8a is the fitted line of *IOW*, and Figure 8b is its corresponding residuals plot. Figure 9a is the fitted line of *K*, and Figure 9b is its corresponding residuals plot. The fitted line of index *IOW* was a/R=0.229IOW+0.0301, the RMSE was 0.01868, and the R-square was 0.9938. Meanwhile, the fitted line of index *K* was a/R=1.772K−0.3176, the RMSE was 0.02234, and the R-square was 0.9911. According to the rule that the value of the R-square is closer to one, the better the result is, it is concluded that the fitting result of *IOW* is better than that of *K*. The RMSE of *IOW* was smaller than that of *K*. Therefore, *IOW* could diagnose crack faults more accurately.

### 4.3. Process of the Improved WNOFRFs

According to the principle of improved WNOFRFs, the process of the improved WNOFRFs for diagnosing crack faults of the rotor system is summarized as follows and shown in Figure 10.

Step 1: Collect the vibration response signals of the crack-free rotor system being subjected to different magnitudes of unbalanced excitations.

Step 2: Calculate the NOFRFs of the crack-free rotor system $G_{n}^{h}(jw),n=1,2,3,4$ through the excitation signals and response signals. Then, calculate its NOFRFs weighted contribution rate $Rn_{h}(n)(\rho ),n=1,2,3,4$ through the following:
(42)$$Rn_{h}(n)=\frac{\int_{-\infty }^{+\infty }\left|T_{n}^{h}(j\omega )\right|d\omega }{\sum_{i=1}^{N}\int_{-\infty }^{+\infty }\left|T_{i}^{h}(j\omega )\right|d\omega }=\frac{\int_{-\infty }^{+\infty }\left|\frac{G_{n}^{h}(j\omega )}{n^{\rho }}\right|d\omega }{\sum_{i=1}^{N}\int_{-\infty }^{+\infty }\left|\frac{G_{i}^{h}(j\omega )}{i^{\rho }}\right|d\omega }\;\;\;\;\;(1\leq n\leq N,\;\rho \in (-\infty ,0)),$$

Step 3: Collect the vibration response signals of the cracked rotor system being subjected to different magnitudes of unbalanced excitations.

Step 4: Calculate the NOFRFs of the inspected rotor system $G_{n}^{t}(jw),n=1,2,3,4$ through the excitation signals and response signals. Then, calculate its NOFRFs weighted contribution rate $Rn_{t}(n)(\rho ),n=1,2,3,4$ through the following:(43)$$Rn_{t}(n)=\frac{\int_{-\infty }^{+\infty }\left|T_{n}^{t}(j\omega )\right|d\omega }{\sum_{i=1}^{N}\int_{-\infty }^{+\infty }\left|T_{i}^{t}(j\omega )\right|d\omega }=\frac{\int_{-\infty }^{+\infty }\left|\frac{G_{n}^{t}(j\omega )}{n^{\rho }}\right|d\omega }{\sum_{i=1}^{N}\int_{-\infty }^{+\infty }\left|\frac{G_{i}^{t}(j\omega )}{i^{\rho }}\right|d\omega }\;\;\;\;\;(1\leq n\leq N,\;\rho \in (-\infty ,0)),$$

Step 5: Calculate the improved WNOFRFs *IW*(*ρ*) through Equation (20).

Step 6: Determine the index *IOW* through Equation (41). Then, conduct fault diagnosis of the inspected system.

## 5. Experiment Study

To evaluate the performance of the fault diagnosis method of improved WNOFRFs, experiments were carried out to diagnose the cracked rotor system. The experimental test bench of the cracked Jeffcott rotor system is illustrated in Figure 11. Different magnitudes of unbalanced forces were implemented by installing different numbers of bolts on the disk. Different degrees of crack faults were implemented by setting different crack depths at the same location of the shaft by a fatigue machine. The depths of the crack faults were 1 mm, 2 mm, 3 mm and 4 mm, as shown in Figure 12. Therefore, the relative depths were 0.2, 0.4, 0.6 and 0.8. Then, the dynamic responses of the experimental rotor system operating at 1600 rpm under various degrees of cracked faults were collected by displacement transducers.

The curves of *IW*1–*IW*4 under different crack faults are illustrated in Figure 13. The improved WNOFRFs (*IW*) were obtained through summing *IW*1–*IW*4. Figure 14 shows the curves of *IW*.

When comparing Figure 13 and Figure 14 with Figure 5 and Figure 6, it was concluded that the performance of the improved WNOFRFs in the experimental study was similar to that in the simulation analysis. It can be seen from Figure 13 that each of *IW*1–*IW*4 rose from 0 to the maximum, which corresponded to the optimal fitness factor. Then, they declined to zero. The peaks of *IW*1–*IW*4 were proportional to the crack depths, and the curves of *IW* had the same trend with the curves of *IW*1–*IW*4. Thus, the changing trend of *IW* in the experiment was also similar to that in the simulation.

Figure 15 shows the values of *K* and *IOW*. It can be seen that the index *K* increased from 0.5594 to 0.9138, and the index *IOW* increased from 0 to 1.9610. Obviously, the index *IOW* was greater than the index *K* when the rotor system was in the same crack fault condition. Therefore, the result of the experiment was consistent with that of the simulation. It can be concluded that the index *IOW* for the crack fault was more sensitive than *K*. Therefore, the method of improved WNOFRFs could effectively diagnose the cracked rotor system.

## 6. Conclusions

A new fault diagnosis method known as improved WNOFRFs was developed by combining the WNOFRFs and KL divergence. A new index, denoted as *IOW*, was proposed to detect a cracked rotor system. The effectiveness of the improved WNOFRFs was proven through the simulation model and experiments of the Jeffcott rotor system. The following conclusions can be drawn:(1)When the relative depth increased from 0 to 0.8, *K* increased from 0.160 to 0.604, and *IOW* went from 0 to 3.410, so *IOW* had the higher sensitivity to crack faults;(2)*IOW* could comprehensively represent the nonlinear properties of the system by integrating all orders of the WNOFRFs;(3)*IOW* could quantitatively represent the crack fault, the minimum relative depth of which could be detected with increments of 0.01;(4)In terms of the simulation model used in this study, the relationship between the values of *IOW* and relative depths of the crack faults was a/R=0.229IOW+0.0301. Therefore, the values of the index *IOW* had an approximately positive proportional relationship with the relative depth of the crack.

Therefore, compared with the original WNOFRFs, which mainly focus on the second weighted contribution rate, the improved method was superior in the fault diagnosis of the cracked rotor system. The index *IOW* manifests many advantages for detecting crack in a rotor system.

Future studies will be focused on the effectiveness of the improved WNOFRFs in identifying other kinds of faults of rotor systems, as well as the relationship between the values of *IOW* and the degree of the fault. Moreover, fault diagnosis of other nonlinear engineering systems will be considered.

## Figures and Tables

**Figure 1 sensors-22-01936-f001:**
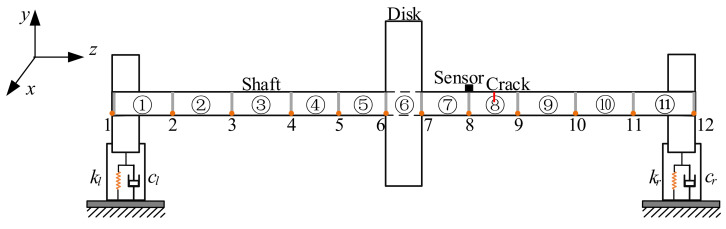
The mathematical model of the cracked rotor system.

**Figure 2 sensors-22-01936-f002:**
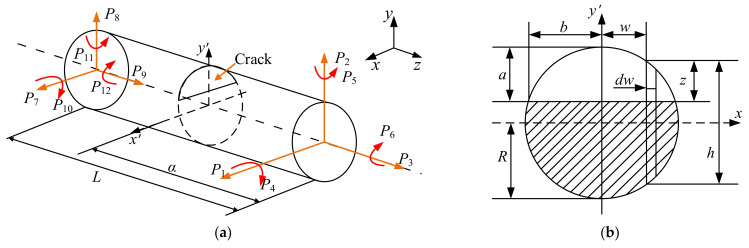
The cracked shaft element. (**a**) A diagram of the cracked shaft element. (**b**) The cross-sectional view of the crack.

**Figure 3 sensors-22-01936-f003:**
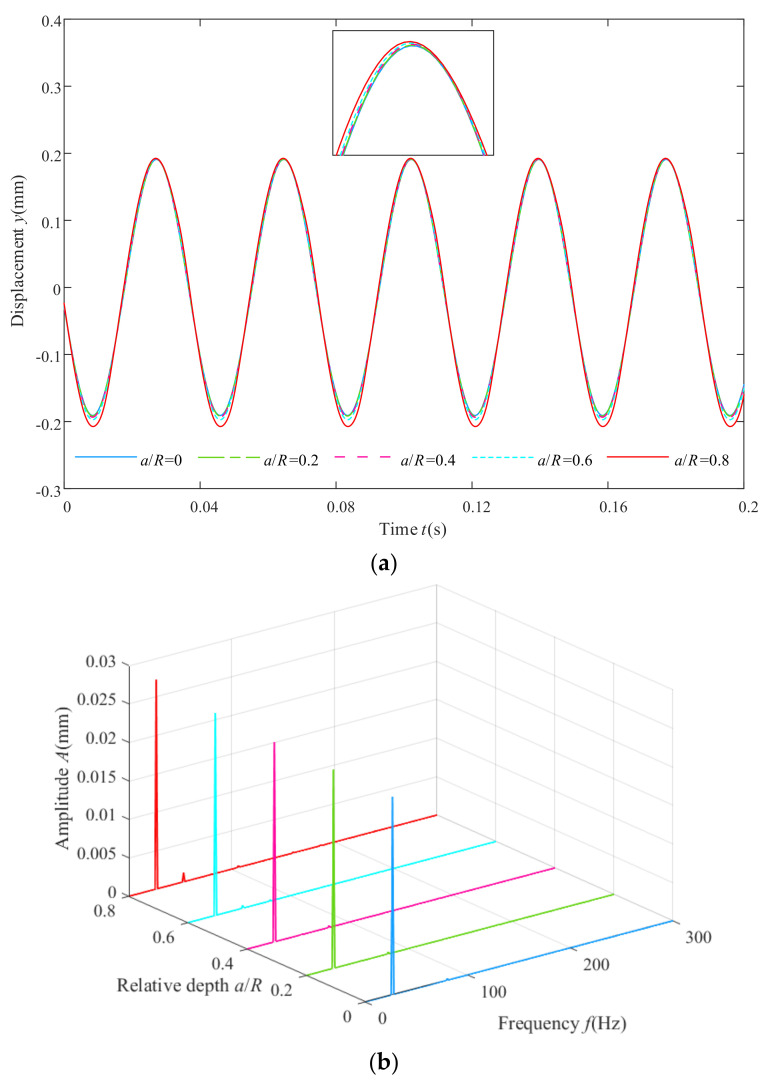
Time domain and frequency domain responses of the Jeffcott rotor system with different crack faults: (**a**) time domain responses and (**b**) frequency domain responses.

**Figure 4 sensors-22-01936-f004:**
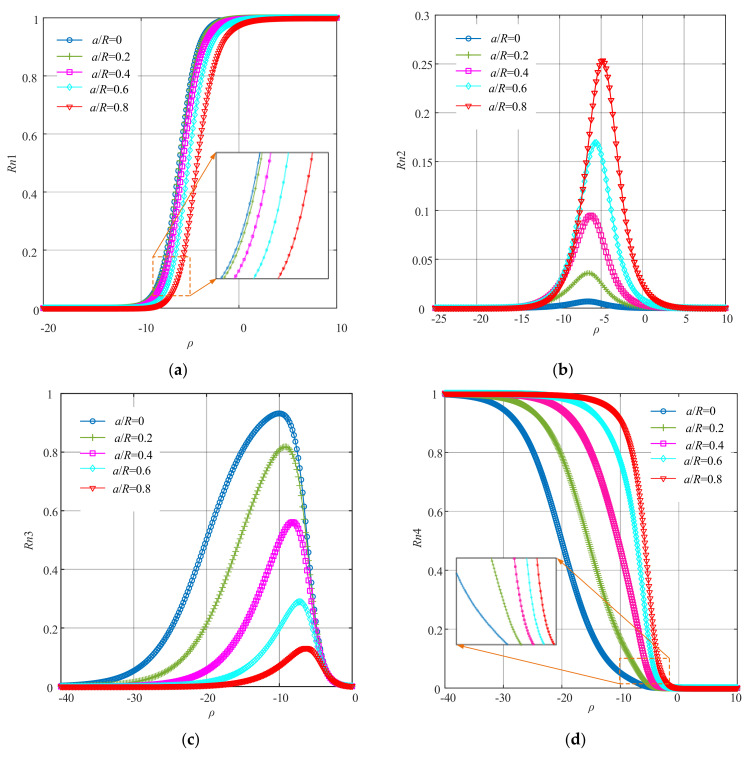
The curves of WNOFRFs under different crack faults: (**a**) *Rn*1, (**b**) *Rn*2, (**c**) *Rn*3 and (**d**) *Rn*4.

**Figure 5 sensors-22-01936-f005:**
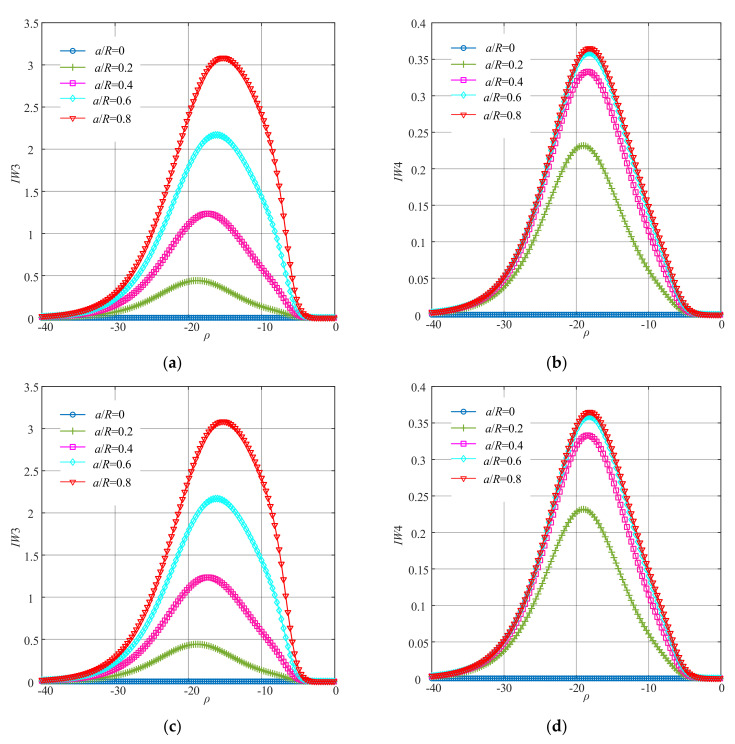
The curves of each order of improved WNOFRFs under different crack faults: (**a**) *IW*1, (**b**) *IW* 2, (**c**) *IW* 3 and (**d**) *IW*4.

**Figure 6 sensors-22-01936-f006:**
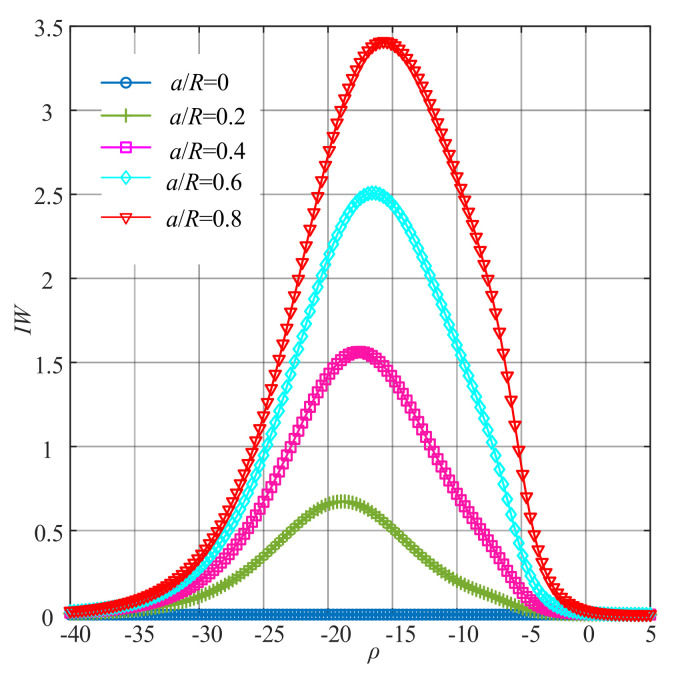
The curves of *IW* under different crack faults.

**Figure 7 sensors-22-01936-f007:**
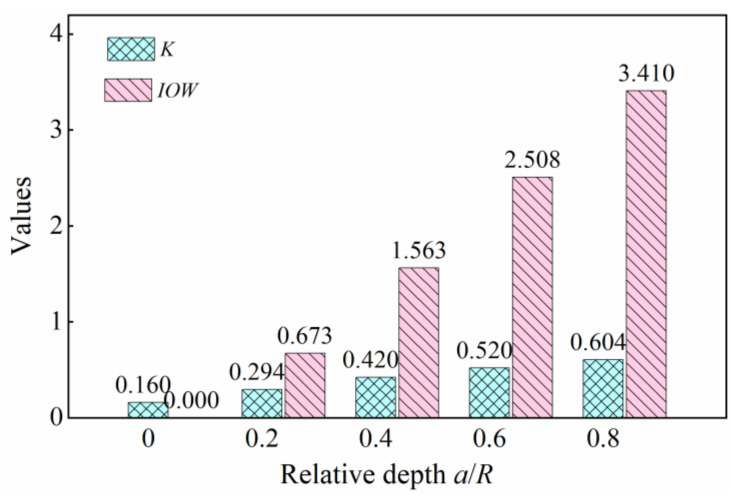
The values of *IOW* and *K*.

**Figure 8 sensors-22-01936-f008:**
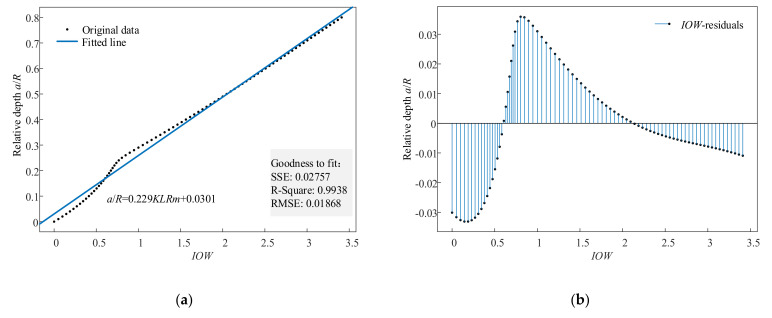
The fitting results of *IOW*: (**a**) fitted line and (**b**) residuals plot.

**Figure 9 sensors-22-01936-f009:**
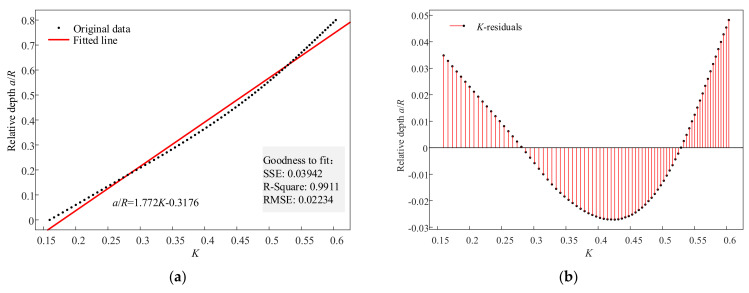
The fitting results of *K*: (**a**) fitted line and (**b**) residuals plot.

**Figure 10 sensors-22-01936-f010:**
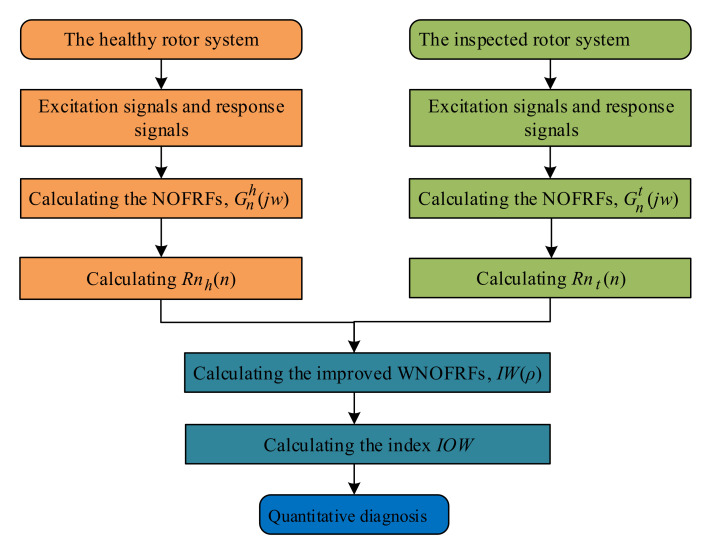
The flow chart of the improved WNOFRFs.

**Figure 11 sensors-22-01936-f011:**
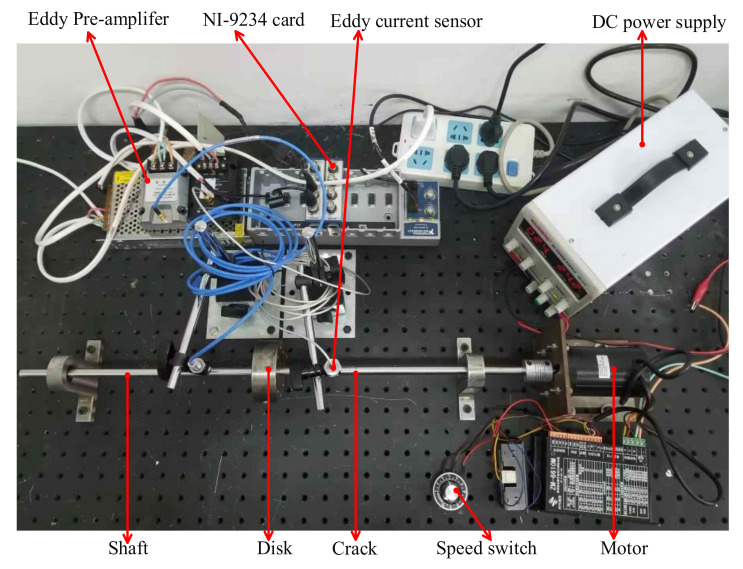
Experimental test bench of the cracked rotor system.

**Figure 12 sensors-22-01936-f012:**
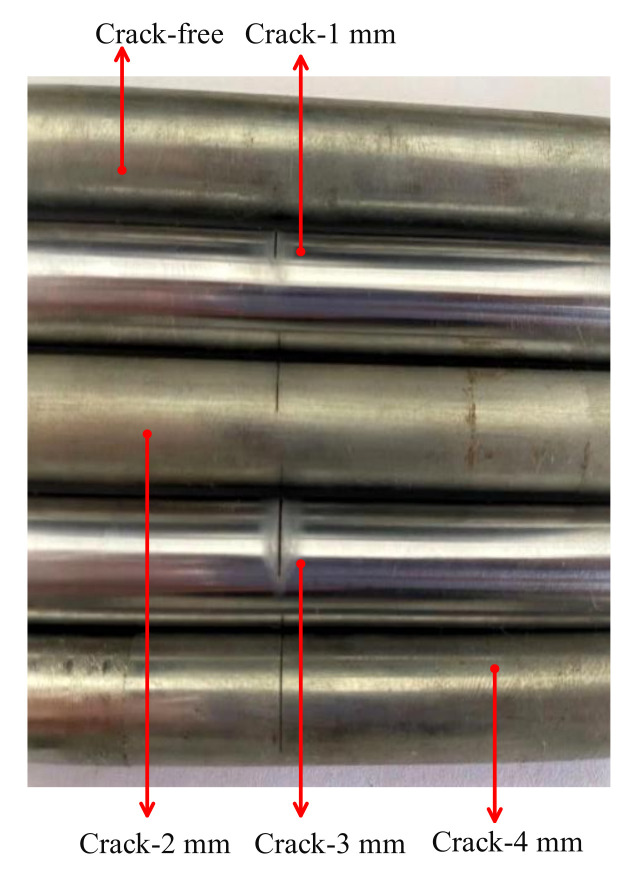
The cracked shafts of the rotor system.

**Figure 13 sensors-22-01936-f013:**
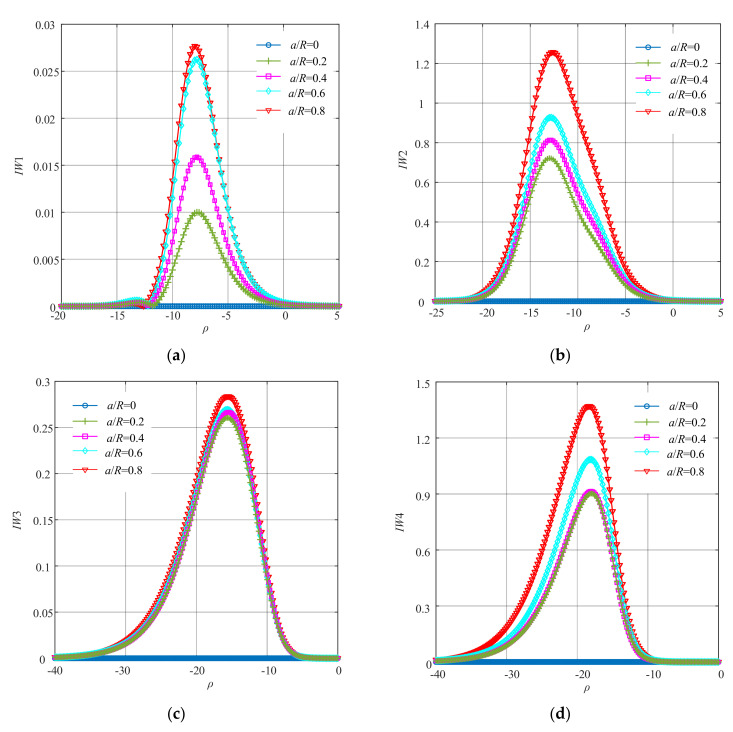
The curves of each order of improved WNOFRFs under different crack faults in the experiment: (**a**) *IW*1, (**b**) *IW*2, (**c**) *IW*3 and (**d**) *IW*4.

**Figure 14 sensors-22-01936-f014:**
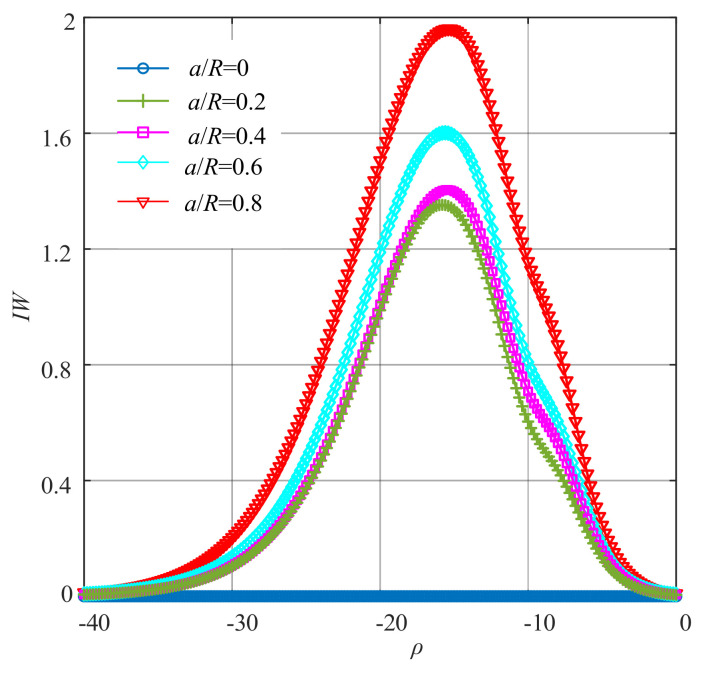
The curves of *IW* under different crack faults in the experiment.

**Figure 15 sensors-22-01936-f015:**
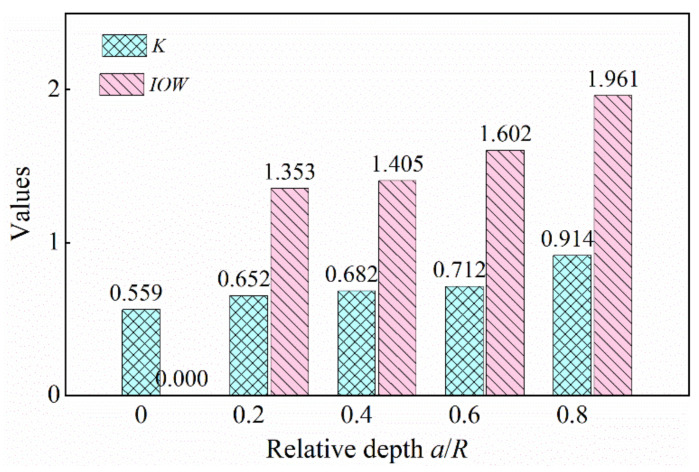
The values of *IOW* and K in the experiment.

**Table 1 sensors-22-01936-t001:** Specific parameters of each unit.

Unit Number	1	2	3	4	5	6	7	8	9	10	11
Radius (mm)	5	5	5	5	5	35	5	5	5	5	5
Length (mm)	50	50	50	40	40	27	40	40	50	50	50

## Data Availability

The data presented in this study are available on a reasonable request from the corresponding author.

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
