# Peer review of "The Improved WNOFRFs Feature Extraction Method and Its Application to Quantitative Diagnosis for Cracked Rotor Systems"

_sensors, 2022, doi:10.3390/s22051936_

Round 1

Reviewer 1 Report

The manuscript entitled "Research on the improved WNOFRFs fault diagnosis method for the cracked rotor system" is a very interesting paper. The authors have provided sufficient literature review in the introduction and describe the idea and methodology of the performed research. Both numerical and experimental studies have demonstrated the effectiveness and advantages of the new index in the diagnosis of the cracked rotor system. The manuscript could be published after some revisions which are suggested as follows.

  1. The flow chart showing methodology should be incorporated in the manuscript.
  2. Captions of Figure 14 and Figure 15 need to be revised to make themselves explanatory. Please check other captions as well. In my opinion captions of different figures should not be the same.
  3. The conclusions should be written in categories or sections.
  4. The use of the space character must be checked, there are many superfluous or missing space characters.
  5. The are some typos. Please check. e.g.:

- 'In Fig. 4(a), the element is loaded' instead of 'In Fig. 4(a), The element is loaded'

- 'its values have a certain relationship' instead of 'its values has certain relationship' (Chapter 2.3)

  1. Whether the proposed method will be affected by the unbalanced force in the detection of cracks? Please explain.
  2. Please explain the meaning of the optimal fitness factor.

Reviewer 2 Report

This manuscript is generally well written. The topic of developing new fault diagnosis method is of interest and has not been explored extensively by other researchers. The methodology adopted/developed by the authors is sound and the conclusions are supported by the analysis and experimental results presented in the paper. With that, I have a few editorial comments as follows.

  1. The acronyms first appearing in the abstract or the main body text should be explained with full spelling of the words. For example, KL and IOW in the abstract; MVA, NARX, CNC, CNN-LSTM, NARMAX, NKL, WNOFRF in the Introduction section.
  2. Line 39, revise “system. And” to “system, and”.
  3. Line 65, revise “have not” to “do not have”.
  4. Line 123, revise “and it” to “, it”.
  5. Line 151, revise “were” to “was”.
  6. Line 171, the reference is missing.
  7. Line 186, revise “And the” to “And if the”.
  8. Line 194 and in several other places, revise “n th” to “nth”.
  9. Line 201, what is the expression for IWN(rho0)?
  10. Line 207, the caption of Table 1 is wrong.
  11. Line 212 and at several other places, add a space between a value and its unit.

Reviewer 3 Report

Paper presents a improved WNOFRFs method for detect the cracked rotor system. In the article, the Authors have pointed which model coefficients may be responsible for the detection of shaft damage. The proposed solution is interesting, but requires a few clarifications and corrections.

In the current configuration, placing the sensor close to the disc and the crack, strongly amplifies the vibration signals. I would ask the Authors to also test and show the results where the sensor will be closer to the bearings (as it is done in most articles). Thanks to this, the bearings will dampen some of the vibrations witch will show how the modification affects the sensitivity of the method. It will give an answer to the possibility of using the method in real conditions.

The shaft diameter is quite small in proportion to its length (the current shaft is flaccid). Can the Authors conduct research for a larger shaft diameter?

Why the tests were performed only at the speed of 1600 rpm. What is the critical speed of the shaft?

Minor errors:

- in the Introduction, many sentences begins with the word “And”. Please rebuild the introduction.

- formulas 10 and 11 should contain indexes

- in 172 lines are Chinese characters

- in formula 20 IWN, N shouldn’t be written in italics

- Chapter 3 describes the rotor model. It is not a new model, but is already known in the literature, why there are no references to literature sources

- lines 433-442 should be removed.

Please also analyze the work Zbigniew Kulesza from 2018 -2020 from journal Shock and Vibrations and Engineering Failure Analysis in which the subject of shaft and turbine diagnostics is discussed based on vibrational signals,  function of correlation and spectral power density. Please also analyze the work of Paolo Pennacchi (2020) from the journal Mechanical Systems and Signal Processing, from 2019 from the Mechanisms and Machine Science journal and from 2018 from the Sensors journal. Also please see work of Mauricio Holguin 2020 from Applied Sciences and Jinde Zheng 2020  from Nonlinear Dynamics.

Reviewer 4 Report

Overview:

The authors present an alternative index for diagnosing a breathing crack in a rotor. They utilize nonlinear frequency response functions. The model utilized in the study is suitable for computing steady state response of rotors. The study could be of interest to engineering community, the authors need to address several queries that will help to enhance the paper prior to publication.

Major comments:

  1. Eq. (40) is identical to Eq. (39). Eq. (40) is supposed to represent harmonic series expansion for time dependent stiffness of the cracked element.
  2. It is unclear how the damping matrix was constructed. What is the amount of modal damping in the system?
  3. What frequency shall the system be excited at? The model is derived to compute the steady state response only and is not suitable for representation of accelerating/decelerating rotors.
  4. The reviewer believes that NOFRFs depend upon the angular position of the unbalance with respect to the crack. This needs to be addressed in the paper.

Minor comments:

  1. Line 14: need to define “IOW”.
  2. Line 76: need to define “NARX”.
  3. Line 81: define “CNN-LSTM”.
  4. Lines 85-91: define indices “Fe”, “Ne”, and “KL” with “NKL”. Define all other indices that are mentioned in the introduction section. If they are not relevant to the paper, it is best not to mention them.
  5. Line 171-172 – something is written in Chinese. Please translate to English.

Please have the paper proofread and edited by professional English editors. Flow of sentences, word usage put a lot of strain on the reader. For example, too many sentences start with word "and".

Round 2

Reviewer 3 Report

The Authors referred to all comments of the reviewer. They conducted additional research in which the sensor was placed close to the bearings. The obtained results were compared with the results for the sensor placed near the disc and cracks. They showed in the coverletter that the method was effective in both variants. I suggest that these results should be included in the article (so that two cases should be considered) - in my opinion, this will raise scientific quality and increase the number of citations of this article.

I recommend the current version of the article for publication in the journal Sensors.